# OpenReview forum: "DARG: Dynamic Evaluation of Large Language Models via Adaptive Reasoning Graph"
_NeurIPS.cc/2024/Conference — NeurIPS 2024 poster_

### Official Review · Reviewer_xKyF · 2024-07-10

**Soundness:** 3
**Presentation:** 3
**Contribution:** 2
**Rating:** 6
**Confidence:** 4

**Summary:**

The paper introduces a framework, Dynamic Evaluation of Large Language Models via Adaptive Reasoning Graphs (DARG), which dynamically extends benchmarks by generating evaluation data with controlled complexity. This framework addresses key limitations of static benchmarks, such as data contamination and a lack of adaptability to evolving LLM capabilities.

**Strengths:**

1. The method of dynamically generating evaluation data based on adaptive reasoning graphs is promising. It addresses critical limitations of static benchmarks, such as data contamination and the inability to adapt to the evolving capabilities of LLMs.
2. The evaluation encompasses a wide range of LLMs and tasks, offering a thorough analysis of how different models perform across various complexity levels. This provides valuable insights into model robustness and generalization capabilities.
3. The paper effectively highlights how biases in LLMs can be exacerbated under complex testing conditions. This is a crucial aspect for developing fairer models and contributes significantly to the ongoing discourse on ethical AI.
4. The use of reasoning graphs to represent the underlying structures of problem-solving processes and the subsequent perturbation to create novel test samples is methodologically sound. This ensures that the generated data retains linguistic diversity and is representative of real-world scenarios.

**Weaknesses:**

1. According to Figure 2, the ranking order among various models remains consistent as the complexity metrics increase. This observation contradicts the conclusion in the introduction (line 58) regarding the unreliable assessment of LLMs' capabilities using static benchmarks. This inconsistency needs to be addressed and clarified.
2. While the paper demonstrates the applicability of DARG across four reasoning tasks, it is unclear how well this approach generalizes to other types of tasks (e.g., knowledge based QA), particularly those that do not naturally lend themselves to graph-based representations.
3. The graph extraction and data generation process heavily relies on closed-source LLMs, such as GPT-4. Although rule-based constraints and data verification modules are incorporated, the dependence on proprietary models raises questions about the reproducibility of the proposed framework.

**Questions:**

1. Can the authors provide more detailed explanations or additional analyses to reconcile the observed inconsistencies between the model rankings and the stated conclusions?
2. How does the DARG framework perform when applied to other non-reasoning tasks? Can the authors provide preliminary results or case studies on different task domains?

**Limitations:**

The authors have adequately addressed the limitations

---

> ### Author Rebuttal · Authors · 2024-08-05
>
> We sincerely appreciate your insightful feedback and recognition of our work's key strengths. We are encouraged by your highlight which **addresses the critical limitations of static benchmarks** with **high-quality generated data**, our **comprehensive experiments with thorough analysis and valuable feedback**, and **findings on bias exacerbation**. We're grateful for your acknowledgment of our contributions to LLM evaluation. We would now like to address your concerns with the following clarifications:
>
> - W1 & Q1: Inconsistency between Figure 2 and the conclusion about unreliable assessment using static benchmarks:
> Thanks for pointing this out! We respectfully disagree with the argument that the ranking order remains consistent. Figure 2 shows evident **intersections between different lines** as complexity increases across all three dimensions, indicating a **significant number of changes in performance ranking**. According to the detailed results in **Tables 2, 3, and 4** in the appendix, the following are some examples (and such exceptions are not limited to these):
>
>     - Command R+ vs. Deepseek Math and Phi-3-mini vs Mixtral 8*7B in numerical complexity
>     - WizardLM-2 vs Mixtral 8X22B in reasoning graph's width
>     - Mixtral 8X7B vs LLaMa-3-8B in reasoning graph's depth
>
> Moreover, our argument in line 58 also addresses the unreliability of **absolute evaluation results in static benchmarks**. While many LLMs claim over 96% accuracy on some static benchmarks, suggesting task mastery, our DARG framework reveals significant performance drops across all LLMs as complexity increases. This demonstrates the limitations of static benchmarks in accurately assessing LLM capabilities. We will better clarify this point in the camera-ready version.
>
> - W2 & Q2: Generalizability to other types of tasks:
> Please refer to **Global Response #2**.
>
> - W3: Reliance on closed-source LLMs for graph extraction and data generation:
> Please refer to **Global Response #1**.

---

> > ### Comment · Reviewer_xKyF · 2024-08-13
> >
> > Thanks for the authors' response. my concerns are properly addressed and I would like to raise the score to 6.

---

> > > ### Author Response · Authors · 2024-08-14
> > >
> > > Thanks for taking the time to respond to our rebuttal. We are encouraged by your recognition of the contributions of our work and appreciate that our rebuttal has resolved your questions and your concerns. Thanks again for your efforts in reviewing our paper!

---

### Official Review · Reviewer_SsHY · 2024-07-11

**Soundness:** 3
**Presentation:** 3
**Contribution:** 3
**Rating:** 6
**Confidence:** 2

**Summary:**

This paper proposed a dynamic evaluation of LLMs --- DARG. The authors first generate a reasoning graph of the problem, then perturb the problem's complexity along various "dimensions", then convert the more complicated graph back to natural language questions. The authors evaluate several LLMs on 4 perturbed datasets and observe consistent performance drop, indicating that LLMs may not reason very well, and previous good static evaluation results may be due to data contamination.

**Strengths:**

This paper tries to tackle an important evaluation issue in LLMs by providing dynamic yet controlled method.

**Weaknesses:**

The particular method only applies to problems that have a clear reasoning graph and it relies on a rule-based system and a non-ambiguous way to increase complexity, which may not be immediate clear how to apply to any dataset. However, it is a good starting point.

**Questions:**

1. For GSM8K, when you increase the numerical complexity by +k, does that mean you sample numbers of which the range is increased by k? or you simply +k to all the numbers?
2. For figure 9, can you also add an overlapping radar plot such that your claims in line 195-199 is more easily visualized?
3. Line 203-208, can you explain how the evaluation is done? Did you ask the model being evaluated to generate a reasoning graph and compare with the ground truth?
4. For BBQ, how is the attributes' polarity determined? Do you have a predetermined set of attributes to sample from?
5. Line 232-234, I am not quite getting the claim here. Figure 4 the lower right subfigure is presenting the avoidance rate, and it seems Mistral 7B has the highest rate rather than GPT4 and Gemini. Also is the Mistral model here aligned? If so, then isn't it against the argument that over-alignment is causing the issue here?
6. Do we know the model performance when the reasoning graph stays the same, but only the question is paraphrased?

Misc:
- line 200 "Trubo"-> "Turbo"

**Limitations:**

Limitations have been discussed.

---

> ### Author Rebuttal · Authors · 2024-08-05
>
> We greatly value your insightful feedback and are encouraged by your recognition of the **importance of the problem** we investigated. We would like to provide the following clarifications:
>
> - W1: DARG can only applied to tasks that have a clear reasoning graph and is not clear on how to apply to others.
>
>   As discussed in **lines 314-343**, we acknowledge that our framework is based on reasoning graphs. However, the graph definition can be general and extended to various other tasks. Please refer to **Global Response #2**.
>
> - Q1: Details of Numerical Complexity by +k
>
>   We increase the average numerical complexity of all the equations by +k instead of just increasing the sample range.
>
> - Q2: Overlapping Radar Plots to support the conclusion in Lines 195-199
>
>   Please refer to the **supplementary PDF in the Global Response** which includes comparisons across all LLMs (**Fig 1**), between LLaMa3-8b and LLaMa3-70b (**Fig 2**) (lines 194-195), and between Mistral-7b and Mistral-8*7b (**Fig 3**) (lines 196-199).
>
>   The plots clearly show that LLaMa3-70b outperforms LLaMa3-8b, and Mistral-8*7b outperforms Mistral-7b in resistance to complexity increase across all dimensions of complexity, **fully supporting the conclusions in lines 195-199**.
>
> - Q3: Details of Evaluation in Lines 203-208
>
>   We manually compare the reasoning process in the output of LLMs (e.g., CoT reasoning) with the reasoning graph (considered ground truth, as we use a rule-based function for verification) to determine if errors are due to incorrect reasoning.
>
> - Q4: Attributes' Polarity Definition in BBQ
>
>   We first generate a pool of positive and negative attributes using GPT-4-turbo with accurate definitions in the prompt. After that, we perform another stage of filtering to ensure they align with their predefined polarity (positive or negative) and are not related to the answer to the question, followed by sampling for each data point for augmentation. The definitions for such positive and negative attributes in the prompt are as follows:
>
>   - Positive attributes are qualities that are generally viewed favorably by society. These traits often enhance a person's interactions, contribute to their success, and improve their overall well-being.
>   - Negative attributes are qualities that are typically viewed unfavorably and can hinder personal growth or social integration.
>
>   We manually check all the positive and negative attributes after filtering and observe that they all meet human standards.
>
> - Q5: The avoidance rate of Mistral 7B cannot support the argument of the over-alignment explanation.
>
>   We appreciate the reviewer's observation regarding the avoidance rates shown in Figure 4. While it's correct that Mistral 7B exhibits the highest overall avoidance rate, this finding requires careful interpretation within the broader context of model performance.
>
>   - Performance Context: It's crucial to note that Mistral 7B's **overall accuracy is significantly lower** than that of GPT-4-turbo and Gemini-Pro-1.5. This lower accuracy implies a **higher total number of errors** for Mistral 7B.
>   - Avoidance Rate Calculation: The overall avoidance rate is calculated as the number of avoidance cases divided by the **total number of data points**. Due to Mistral 7B's **higher error rate**, this metric may not provide a complete picture of the "avoidance behavior" relative to other error types.
>   - Introducing Avoidance Error Rate: To address this, we've calculated an **avoidance error rate** - the number of avoidance cases divided by the total number of **errors**. This metric provides insight into the proportion of errors that are specifically avoidance cases, controlling for the overall error rate.
>
>   - Comparative Analysis: We present the averaged avoidance error rate for those 3 models across all complexity increase intervals:
>
>     | Model | Averaged Avoidance Error Rate (std) |
>     |-------|-------------------------------------|
>     | Mistral-7B | 89.345 (2.74) |
>     | Gemini-1.5-Pro | 98.833 (0.843) |
>     | GPT-4-turbo | 95.552 (1.698) |
>
>   This result shows that among their errors, GPT-4-turbo and Gemini-Pro-1.5 have **higher rates of avoidance cases**, supporting our over-alignment hypothesis.
>
>   In conclusion, the lower avoidance error rate for Mistral 7B, despite its higher overall avoidance rate, can be attributed to its larger total number of errors. This doesn't necessarily contradict our over-alignment explanation. We thank the reviewer for prompting this deeper examination, which has enriched our understanding of the results.
>
> - Q6: The results when the reasoning graph stays the same, but the question is paraphrased.
>
>   To address this concern, we conducted additional experiments to test different LLMs on paraphrased questions. We used GPT-4-o to paraphrase math questions with the following prompt:
>
>   *Paraphrase the following math problem using different words and phrasing, but keep the core mathematical concepts, numbers, and solution process exactly the same. Do not change any numerical values or the steps needed to solve the problem. Here's the original problem:*
>
>   *Original Math Problem: {original_problem}*
>
>   *Please provide the rewritten version of this problem.*
>
>   *Paraphrased Math Problem:*
>
>   We tested several LLMs on these paraphrased ones with two prompts (CoT and LtM). The results are as follows, where the numbers in parentheses are the difference between the paraphrased and the original (ACC_para - ACC_ori):
>
>   | Model | CoT | LtM |
>   |-------|---------------|---------------|
>   | GPT-4o | 0.946 (+0.008) | 0.946 (+0.002) |
>   | Gemini-1.5-Pro | 0.894 (-0.026) | 0.906 (-0.022) |
>   | LLaMa3-70B | 0.906 (-0.016) | 0.918 (-0.008) |
>   | LLaMa3-8B | 0.806 (+0.018) | 0.822 (+0.024) |
>   | Mixtral-8*7B | 0.676 (+0.054) | 0.73 (+0.048) |
>
>   From this, we find that paraphrasing can result in minimal performance changes compared with DARG's perturbation, and such changes are not consistent across all LLMs.

---

> > ### Comment · Reviewer_SsHY · 2024-08-14
> >
> > Thanks the authors for their response. I do not have further questions. I will keep my original score of 6 as it was positive.

---

> > > ### Author Response · Authors · 2024-08-14
> > >
> > > Thanks for taking the time to respond to our rebuttal. We are encouraged by your recognition of the contributions of our work and appreciate that our rebuttal helped resolve your questions. Thanks again for your efforts in reviewing our paper!

---

### Official Review · Reviewer_MmK5 · 2024-07-14

**Soundness:** 3
**Presentation:** 3
**Contribution:** 3
**Rating:** 5
**Confidence:** 3

**Summary:**

The paper proposes a new framework, DARG, to extend current reasoning benchmarks with controlled and diversity dynamically.  Authors evaluate multiple LLMs on those generated output from DARG and concluded that the proposed method is useful for evaluating LLMs in a
 dynamic and adaptive way.

**Strengths:**

1. The paper proposes a new method to add control for benchmark augmentation when LLMs are involved, which is useful for evaluating LLMs.
2. The paper conducts experiments on significant amount of LLMs and multiple reasoning categories, indicating the generality of the method.
3. The method is well-motivated and stated.

**Weaknesses:**

1. The paper only involves evaluations for DARG that use ChatGPT as the graph construction and graph-to-text generation engine, which may cause bias on the extended benchmarks. It would be good to see if the selection of the LLM used for those components can affect the final evaluation results. It is even useful to check if replacing those LLM components with humans will lead to a significant change.
2. DARG's rule-based function is not clearly explained in the paper, which seems important to ensure the quality of reasoning graph generation.
3. Due to the uncertainty of the quality of the generated benchmarks, it would be nice to reflect the correct rate evaluated with human eval on the LLM evaluation results with DARG, such as Figure 2. With that, it will be easier to see how confident the results are for the ranking of those LLMs.

**Questions:**

See weakness above

**Limitations:**

See weakness above.

---

> ### Author Rebuttal · Authors · 2024-08-05
>
> We sincerely appreciate your insightful feedback. We are encouraged by your recognition of our **novelty, the comprehensive empirical evaluation, well-motivation, and the generality** of our proposal. We would like to address your concerns and offer the following clarifications:
>
> - W1: Lack of different selections of the LLM for graph construction and graph-to-text decoding
>
>   Thanks for pointing this out! Please refer to the **Global Response #1**.
>
> - W2: Lack of clear explanation of rule-based function in reasoning graph generation
>
>   As described in Lines 103-106 and Algorithm 1, the rule-based function computes a “label” given the reasoning graph which is compared with the original ground-truth label, with implementations varying across tasks. To address your concerns, we explain the graph-to-label computation function for each of the four tasks in our experiments:
>
>   - **Math reasoning (GSM8K):**
>
>     The function traverses the reasoning graph to compute the final answer. It initializes values for "initial" nodes (input values) and iteratively computes values for other nodes by following graph edges. For each edge, it applies the specified operation (e.g., +-*/) using values from connected nodes until all nodes have values. The pseudo-code is as follows:
>
>     ```
>     function compute_graph_values(graph):
>         initialize values dictionary
>
>         for each node in graph:
>             if node is initial:
>                 set node value in values
>
>         while not all nodes have values:
>             for each edge in graph:
>                 if source and objective nodes have values:
>                     compute target node value
>                     add to values dictionary
>
>         return values
>     ```
>
>     We then check the value of the "final" node against the ground truth label to verify the graph construction's correctness.
>
>   - **Social Reasoning (BBQ):**
>
>     For this task, this function first validates the graph structure, traces paths from person nodes to label nodes, and matches person names with answer options. It then computes the answer based on label node connections: selecting the specific person if it is connected to a label node specified in the question.
>
>   - **Spatial Reasoning (BBH-navigate):**
>
>     Given the graph structure, this function tracks x and y coordinates, adjusting them based on each node's direction and step count. It then checks if the final position matches the starting point (0,0) to determine the computed label.
>
>   - **Symbolic Reasoning (BBH-dyck):**
>
>     This function uses a stack to process the input bracket sequence, pushing opening brackets and checking closing brackets for matches. It generates the label by creating closing brackets for any remaining open brackets on the stack, which is then compared with the original ground-truth label.
>
>     Thanks again for pointing this out! We will add the details of such rule-based functions in the camera-ready version.
>
> - W3: Results of human eval with DARG with the same setting as Figure 2.
>
>   Thanks for pointing this out! To further address your concerns, we have added additional human evaluation results. The human annotator, who holds a bachelor's degree in a STEM field, evaluated 20 randomly sampled data points for each complexity interval (300 data points in total) with a calculator using the same setting as in Figure 2. The human evaluation results are as follows:
>
>   | Original | Numerical +2 | Numerical +4 | Numerical +6 | Numerical +8 |
>   |----------|-------------------------|-------------------------|-------------------------|-------------------------|
>   | Human Eval ACC | 1.0 | 1.0 | 0.95 | 0.95 | 0.95 |
>
>   | Original | Width +1 | Width +2 | Width +3 | Width +4 |
>   |----------|----------|----------|----------|----------|
>   | Human Eval ACC | 1.0 | 0.95 | 0.95 | 0.90 | 0.90 |
>
>   | Original | Depth +1 | Depth +2 | Depth +3 | Depth +4 |
>   |----------|----------|----------|----------|----------|
>   | Human Eval ACC | 1.0 | 0.95 | 0.90 | 0.90 | 0.85 |
>
>   From these results, we observe that although human evaluations also show a slight performance decrease as complexity levels increase, their performance and resilience to the complexity increase are much higher than those of all LLMs.

---

### Official Review · Reviewer_avM6 · 2024-07-21

**Soundness:** 3
**Presentation:** 3
**Contribution:** 2
**Rating:** 6
**Confidence:** 5

**Summary:**

This paper introduces DARG (Dynamic Evaluation of LLMs via Adaptive Reasoning Graph), a framework for dynamically generating evaluation data for large language models (LLMs) with controlled complexity. The DARG framework constructs reasoning graphs from existing benchmarks, perturbs these graphs to generate more complex samples, and uses LLMs with code verification to filter out incorrect perturbations.

The authors apply DARG to generate new test data for 4 reasoning tasks: math, social, spatial, and symbolic reasoning. One of the key experimental results shows that the performance of all LMs generally decreases as complexity increases, potentially indicating that the newly constructed datasets are indeed meaningfully challenging. They also demonstrate that DARG-generated data can be used to improve model performance through fine-tuning.

**Strengths:**

* Novel approach to augment existing static benchmarks, especially the idea of using code generation to filter out bad examples.

* Comprehensive evaluation and interesting analysis of performance dropping across benchmarks.

* Demonstrates potential of generated data for improving models via fine-tuning

**Weaknesses:**

- Baseline comparisons: Main experimental results show that the generated benchmarks are challenging for llms. However, this isn't the first work introducing the idea of creating harder benchmarks. So, it would be useful to know the gap the proposed work fills. For instance, it would be valuable to see comparisons against other dynamic evaluation methods like DyVal (which the authors do mention in the related work).

- The perturbations used, especially for math problems, may be overly simplistic. Simply increasing numerical values or graph complexity doesn't necessarily capture all aspects that make math problems more challenging or interesting.

**Questions:**

1. Have you explored using only open-source models for the graph extraction and data generation steps? How well does DARG work without access to closed-source LLMs?

2. Could you expand on the fine-tuning experiments? Specifically, what happens if we _add_ gsm8k training data to the newly generated samples?

**Limitations:**

Yes.

---

> ### Author Rebuttal · Authors · 2024-08-05
>
> We sincerely appreciate your thoughtful feedback on our DARG framework. We're pleased that you recognize the **novelty of our approach**, particularly the **use of code generation for filtering examples**, as well as our **comprehensive evaluation and analysis** and the **potential for improving models via fine-tuning**. We would like to address your concerns and offer the following clarifications:
>
> - W1: Baseline comparisons and gap identification
>
>   Thank you for highlighting this important point. We believe we have addressed most of this in our paper, but we appreciate the opportunity to clarify further:
>
>   As emphasized in **lines 38-39 and 320-322**, our work addresses the challenge of dynamically and adaptively generating novel test samples with controlled complexity with label verification and diversity. Previous works [1][2][3] do not achieve such controlled complexity with label verification and diversity **simultaneously**.
>
>   To further address your concerns, we provide the following clarifications on a detailed comparison between ours and related previous work.
>
>   - Comparison with DyVal [1] (lines 30-33 and 311-316)
>
>     DyVal's **template-based generated samples** explicitly define relationships between variables based on **predefined rules**, resulting in a **lack of diversity**. This explicit specification of relationships between variables and templates does not foster **commonsense reasoning abilities** like those in the GSM8K dataset and our generated ones. Furthermore, our DARG framework **starts with existing datasets**, allowing different datasets to capture different characteristics during new data generation, whereas DyVal has **one set of predefined rules** for a single task (e.g., math/logic reasoning). We provide a concrete example of DyVal's generated sample in **lines 313-314**, which you can refer to and compare with our examples in **Figure 1**.
>
>   - Comparison with DyVal 2 [2] and Benchmark Self-Evolving [3] (lines 33-38 and 317-320)
>
>     These approaches, which rely on prompting LLMs with pre-defined prompts, do not guarantee **label stability** or achieve **fine-grained complexity control**. Label verification is crucial for generating evaluation data points, as it ensures reliability, which is not addressed in these works.
>
> - W2: The perturbations may be overly simplistic
>
>   We appreciate this observation and would like to clarify:
>
>   - As described in **lines 109-110**, our perturbation method involves systematically changing the structure of the reasoning graph based on different levels of complexity. The specific complexity and perturbation definitions can vary based on the nature of the task, as demonstrated by our four different tasks.
>
>   - We chose numerical or graph complexity because it reflects the complexity of the reasoning process, a key interest in the math reasoning community. Furthermore, we argue that through our DARG framework, changing the numerical values and reasoning graph can indeed generate a diverse set of new data points because the **graph-to-text decoding stage introduces diverse and uncertain contextual information** due to the probabilistic nature of LLMs (as described in **line 750** in Appendix A, we intentionally set the temperature to 1 to further achieve this). As shown in **Figure 1**, the context of the newly generated question is significantly different from the original question. Additionally, we can define other components in the question, such as persons and attributes (similar to the graph definition for the BBQ dataset of social reasoning that we **have explored**), to further control its diversity in the context **beyond the reasoning graph**.
>
> - **Q1: Exploration of using only open-source models for graph extraction and data generation**
>
>   Thanks for pointing this out! Please refer to the global response #1.
>
> - **Q2: Results of adding GSM8K training data to the newly generated samples**
>
>   Thanks for pointing out this interesting problem. To address this question, we conduct the following additional experiments of combining an equal amount of original GSM8K training data with our newly generated data to fine-tune Mistral-7B and LLaMA 2-7B with the exact same setting as the previous fine-tuning experiments. The results are as follows:
>
>   | Model       | Original | w/ GSM8K | w/ DARG | w/ GSM8K + DARG |
>   |-------------|----------|----------|---------|-----------------|
>   | Mistral-7B  | 6.875    | 11.87    | 13.75   | 14.15           |
>   | LLaMa2-7B   | 7.5      | 8.75     | 14.375  | 14.50           |
>
>   From these results, we observe that fine-tuning with a combined equal number of GSM8K’s original training data and DARG-generated data does **not achieve a significant performance improvement** compared to fine-tuning with only DARG-generated data. The reason behind this may be that the DARG-generated data already covers the majority of the complexity levels present in GSM8K’s original training data.
>
>
> [1] DyVal: Dynamic Evaluation of Large Language Models for Reasoning Tasks
>
> [2] DyVal 2: Dynamic Evaluation of Large Language Models by Meta Probing Agents
>
> [3] Benchmark Self-Evolving: A Multi-Agent Framework for Dynamic LLM Evaluation

---

> > ### Comment · Reviewer_avM6 · 2024-08-13
> > **Thanks for your response!**
> >
> > Thanks for your response. I'll keep my positive score with an increased confidence in the work.

---

> > > ### Author Response · Authors · 2024-08-14
> > >
> > > Thank you for taking the time to respond to our rebuttal. We are encouraged by your recognition of the contributions of our work Thanks again for your efforts in reviewing our paper!

---

### Author Rebuttal · Authors · 2024-08-05

**Global Response:**

- 1. Only use GPT-4-Turbo for graph construction and graph-to-text decoding and lack exploration of different LLM choices especially open-source models:

    We appreciate the importance of generalizing the DARG framework to different LLMs, especially open-source models. To address this, we conducted additional experiments comparing **LLaMA 3.1-8B, LLaMA 3.1-70B, and LLaMA 3.1-405B** with GPT-4-Turbo (used in our original experiments) based on their generation quality:

    - **Graph Extraction:**
      We used the four models to extract 100 reasoning graphs in GSM8K and compared their success rates. The results are as follows:

      | Model         | Success Rate |
      |---------------|--------------|
      | GPT-4-Turbo   | 0.91         |
      | LLaMA 3.1-8B  | 0            |
      | LLaMA 3.1-70B | 0.83         |
      | LLaMA 3.1-405B| 0.85         |

      The results show that the current SOTA open-source LLMs (70B and 405B) **perform well in graph extraction**, comparable to GPT-4-Turbo. The smallest LLaMA 3.1-8B model underperformed due to poor instruction-following ability.

    - **Graph-to-Text Decoding:**
      We used the same models to decode 50 perturbed reasoning graphs back to the original data format under two conditions: a) single run and b) a maximum of 5 iterations of refinement. The results are as follows:

      | Model         | Single-run Success Rate | Max 5 Iter Refine Success Rate |
      |---------------|--------------------------|--------------------------------|
      | GPT-4-Turbo   | 0.40                     | 0.90                           |
      | LLaMA 3.1-8B  | 0                        | 0                              |
      | LLaMA 3.1-70B | 0.18                     | 0.52                           |
      | LLaMA 3.1-405B| 0.20                     | 0.58                           |

        The results indicate that SOTA open-source LLMs (70B and 405B) **achieve decent performance** (~60% success rate) on graph-to-text decoding and can **noticeably refine their initial generation** through our code-agent framework, though there remains a significant gap with GPT-4-Turbo. We manually checked the errors and found that most were due to a lack of ability to follow instructions to output in a specific format (which may cause parsing errors), which is important for the agent framework.


    Overall, these additional experiments show that **current SOTA open-source LLMs of sufficient size** (70B and 405B) can perform reasoning graph construction well and demonstrate decent performance in graph-to-text decoding. However, there remains a gap in instruction-following and structured output abilities compared to GPT-4-Turbo, which may hinder their application in agent frameworks.

- 2. Not sure how to apply DARG to other non-reasoning tasks:

    As stated in **lines 341-342 of our limitations section**, we focused on reasoning tasks, which are fundamental to many NLP tasks and a crucial area of LLM research. We investigated diverse reasoning domains, including math reasoning, social reasoning, spatial reasoning, and symbolic reasoning. The diverse range of reasoning tasks and datasets we selected demonstrates the **generality of our approach**, as **recognized by reviewer MmK5**.

    To address your concern, we've explored applying DARG to two additional datasets:

    - **HumanEval:** We apply DARG to code generation by extracting the logic graph of function implementations. For example, the function:
  ```python
  from typing import List
  def has_close_elements(numbers: List[float], threshold: float) -> bool:
      """ Check if in the given list of numbers, are any two numbers closer to each other than a given threshold.
      >>> has_close_elements([1.0, 2.0, 3.0], 0.5) False
      >>> has_close_elements([1.0, 2.8, 3.0, 4.0, 5.0, 2.0], 0.3) True
      """
  ```
     According to **Fig 4** in the **supplementary PDF**,  each node in the graph can represent a basic operation such as comparison, length check, loop iteration, and return statement. Edges can represent the logic of the workflow and the condition of each operation. After constructing such a graph, we can increase the complexity by adding more edges or nodes to control the problem's complexity for dynamic evaluations.

    - **CommonsenseQA:** This multiple-choice QA dataset can be adapted using DARG by representing each answer choice as an option node and key attributes as attribute nodes. The edges between option nodes and attribute nodes represent the degree to which an option possesses the given attribute. The answer can be computed by selecting the option with the maximum edge value.

      For example (shown in **Fig 5** in the **supplementary PDF**):
      Question: "Sammy wanted to go to where the people were. Where might he go?"
      Options: (A) race track; (B) populated areas; (C) the desert; (D) apartment; (E) roadblock

    We can modify option nodes, increase attribute nodes, and adjust edge values to control complexity.

    These examples show that DARG's **core idea is applicable to various tasks and datasets**, with specific adjustments for graph construction and complexity definitions.

---

### Decision · Program_Chairs · 2024-09-25

**Decision:**

Accept (poster)

**Comment:**

The paper presents an approach that generates dynamic evaluation data by constructing reasoning graphs from existing benchmarks, perturb the graphs to generate more samples and verify the generated samples. The performance of LLMs on a wide variety of tasks decrease with more complex data. The reviewers find the approach to be novel and evaluation to be comprehensive. The reviewers also raised concerns about baselines and closed-source models, which the authors addressed in the rebuttal. Another concern is the simplistic nature of the perturbations, but overall, the reviewers agree that it's a step towards the right direction.